# Effect of Hydrochloric Acid Hydrolysis under Sonication and Hydrothermal Process to Produce Cellulose Nanocrystals from Oil Palm Empty Fruit Bunch (OPEFB)

**DOI:** 10.3390/polym16131866

**Published:** 2024-06-29

**Authors:** Zulnazri Zulnazri, Rozanna Dewi, Agam Muarif, Ahmad Fikri, Herman Fithra, Achmad Roesyadi, Hanny F. Sangian, Sagir Alva

**Affiliations:** 1Chemical Engineering Department, Malikussaleh University, Lhokseumawe 24353, Aceh, Indonesia; rozanna.dewi@unimal.ac.id (R.D.); amuarif@unimal.ac.id (A.M.); 2Center of Excellence Technology Natural Polymer and Recycle Plastics, Malikussaleh University, Lhokseumawe 24353, Aceh, Indonesia; ahmadfikri@unimal.ac.id; 3Material Engineering Department, Malikussaleh University, Lhokseumawe 24353, Aceh, Indonesia; 4Civil Engineering Department, Malikussaleh University, Lhokseumawe 24353, Aceh, Indonesia; hfithra@unimal.ac.id; 5Chemical Engineering Department, Sepuluh Nopember Institute of Technology, Surabaya 60111, Indonesian, Indonesia; aroesyadi@yahoo.com; 6Physics Department, Sam Ratulangi University, Manado 95115, Indonesian, Indonesia; hannysangian@yahoo.co.id; 7Mechanical Engineering Department, Mercu Buana University, Jakarta 11650, Indonesia; sagir.alva@mercubuana.ac.id

**Keywords:** cellulose, cellulose nanocrystals, hydrolysis

## Abstract

This paper presents an approach for hydrolyzing cellulose nanocrystals from oil palm empty fruit bunch (OPEFB) presented through hydrochloric acid hydrolysis under sonication–hydrothermal conditions. Differences in concentration, reaction time, and acid-to-cellulose ratio affect toward the yield, crystallinity, microstructure, and thermal stability were obtained. The highest yield of cellulose nanocrystals up to 74.82%, crystallinity up to 78.59%, and a maximum degradation temperature (T_max_) of 339.82 °C were achieved through hydrolysis using 3 M HCl at 110 °C during 1 h. X-ray diffraction analysis indicated a higher diffraction peak pattern at 2θ = 22.6° and a low diffraction peak pattern at 2θ = 18°. All cellulose nanocrystals showed a crystalline size of under 1 nm, and it was indicated that the sonication–hydrothermal process could reduce the crystalline size of cellulose. Infrared spectroscopy analysis showed that a deletion of lignin and hemicellulose was demonstrated in the spectrum. Cellulose nanocrystal morphology showed a more compact structure and well-ordered surface arrangement than cellulose. Cellulose nanocrystals also had good thermal stability, as a high maximum degradation temperature was indicated, where CNC-D1 began degrading at temperatures (T_0_) of 307.09 °C and decomposed (T_max_) at 340.56 °C.

## 1. Introduction

Cellulose nanocrystals (CNC) are nano-sized particles, have a regular atomic structure and high crystallinity, and do not have amorphous regions [1,2]. These nanoparticles have received much attention from scientists because of their exceptional mechanical properties, such as a 138 Gpa Young’s modulus and a 1.7 GPa tensile strength. On the other hand, because of its high surface reactivity, CNC materials have been successfully used as an organic filler material for biopolymer nanocomposites [3,4,5]. Moreover, CNC applications in the medical field can be employed as an ingredient in medicine manufacturing [6].

CNCs were produced from cellulose with a two-step process. The initial hydrolysis was to remove amorphous regions of the cellulose polymer, and then the process continued with the fragmentation of crystal segments to generate nanocrystals [7]. Hydrolysis was the method employed for CNC production, using 60–64% of sulfuric acid solutions at 50 °C, followed by fragmentation and neutralization to produce crystalline cellulose [8,9,10,11]. Unfortunately, the hydrolysis process using sulfuric acid solutions had weaknesses, such as the yield obtained being less than 35% [12,13] and degradation of crystalline cellulose chains. The product obtained also showed low thermal stability with a maximum degradation temperature (Tmax) of 250 °C [14]. The CNC degradation originated from a dehydration reaction caused by the residual sulfuric (SO_4_^2−^) group on the material. Although the sulfuric group could be removed from the substrate with dialysis or desulphurized, the technique was complicated and avoiding particle aggregation was challenging [15,16,17]. Therefore, selecting a suitable material for hydrolysis is crucial to preventing accumulation and producing high crystallinity.

CNCs were produced from renewable and abundant biomass such as corn fiber, coconut husk, banana peel, bagasse, and empty oil palm bunches. Oil palm empty fruit bunches (OPEFBs) are lignocellulosic waste that was not utilized optimally. They were only used as raw material for compost, boiler fuel, and road hardener in plantations. Apart from the abundant amount, OPEFBs also contain very high cellulose fibers of up to 64%; the other contents are hemicellulose and lignin [18]. Cellulose is a semi-crystalline polysaccharide with a D-glucopyranosyl unit connected with B-(1-4)-glucosidic bonds [19]. Cellulose is soluble in acid solutions but insoluble in alkaline solutions, whereas both hemicellulose and lignin are easy to dissolve in alkaline solutions but insoluble in acid solutions. By employing alkaline solutions in pretreatment, hemicellulose and lignin could be removed from cellulose [20].

Several low concentrations of mineral acids, such as hydrochloric acid and maleic acid under hydrothermal and ultrasonic conditions, were used to prevent aggregation and crystalline degradation [15,21,22,23]. Authors Pawcenis and Rubleva reported that CNC substrates produced from hydrolysis introduced hydrochloric acid solutions that showed high thermal stability [24,25]. The process provided flocculated suspension and generated a low yield recorded at 20%. The method, which applied a high-power ultrasonic irradiation with a short sonication period, obtained approximately 2–5% yield [16]. Tang found that the hydrolysis production of cation-exchange resins produced a yield of 50.04%. However, removing cation-exchange resins using post-treatment and continuous centrifuge took longer and required chemical solvents [26].

Hydrolysis of pulp using 6 M HCl under hydrothermal conditions resulted in 80.80% of products and a 35–49 nm size distribution [23]. The authors also investigated the high product yield being influenced by weak acid with a shorter processing time, in which acid penetrated the inner layer of cellulose tissue rapidly and hydrolyzed the amorphous region of cellulose chains. Meanwhile, crystalline parts of cellulose were more resistant to hydrolysis using weak acid because the hydrogen bond strength between cellulose is higher than the amorphous area, which were less compact. Thus, the time of reaction and acid concentrations were the main parameters for hydrolyzing cellulose to CNCs.

The same approach was adopted in this work to obtain the CNC product from OPEFBs, which were extracted to gain cellulose. Then, they were hydrolyzed using low concentrations of hydrochloric acid solutions under sonication and hydrothermal conditions. This work emphasized optimizing the process with insufficient reaction time and acid concentrations to produce a high-crystallinity and high-yield CNC product.

## 2. Materials and Methods

### 2.1. Materials

Oil palm fruit bunches, which are biomass residue, were taken from a crude palm oil processing plant. The chemical substance used for OPEFB extraction was NaOH 17.5% from Merck, Germany and NaOCl 2% was purchased from PT. Bratachem, Surabaya, Indonesia. The acid solution used for cellulose hydrolysis was 1, 2, 3, 4, and 5 M HCl from Merck, Germany. Aquadest, which was used for washing the samples, was taken from PT. Bratachem, Surabaya, Indonesia.

### 2.2. Instrumentation

The instruments employed in this work were as follows: an Ultrasonic bath of type SU-27 TH with a capacity of 477(W) × 272(D) × 200(H) mm From ULTRATECNO British, a hydrothermal pressure batch reactor from Parr USA, a Nicolet 8700 Fourier transform infrared (FT-IR) spectrophotometer, an Oxford INCA 400 scanning electron microscopy (SEM) with a voltage of 15 KV from Oxford INCA British, a Philips PZ1200 X-ray diffractometer (X-RD) from Malvern PANalytical United Kingdom, and a NETZSCH TG 209 F1 thermogravimetric analyzer (TGA).

### 2.3. CNC Preparation

Fifty grams of dry-weight OPEFBs were included in the 500 mL of 17.5 wt% NaOH (1:10) *w*/*v*, and then the sample was refluxed at 80 °C for 2 h. Then, the sample was filtered, and the residue was collected as a cellulose extract. Further bleaching was performed with 250 mL of 2 wt% NaOCl at 70 °C for 1 h. Then, the cellulose extract obtained was filtered and washed until neutral.

Cellulose 1:60 HCl (g/mL) was inserted in the SU-27 TH Ultrasonic bath with a capacity of 477(W) × 272(D) × 200(H) mm, frequency of 28 kHz, output of 300 watts, and heater of 500 watts during 30 min. After that, the sample was hydrolyzed in the hydrothermal Parr USA pressure batch reactor.

The reaction was run through several variables: at a temperature of 110 °C, hydrolysis with 1 M HCl for 30 min starting from a temperature of 25 °C until it reached 110 °C for CNC-A0, hydrolysis with 2 M HCl for 30 min starting from a temperature of 25 °C until reaching a temperature of 110 °C for CNC-B0, and so on for CNC-C0, CNC-D0, and CNC-E0; the hydrolysis process with 1 M HCl with a reaction time of 1 h at a temperature of 110 °C for CNC-A1, the hydrolysis process with 2 M HCl with a reaction time of 1 h at a temperature of 110 °C for CNC-B1, the hydrolysis process with 3 M HCl with a reaction time of 1 h at a temperature of 110 °C for CNC-C1, and so on for CNC-D1 and CNC-E1; for a temperature of 120 °C, hydrolysis using 3 M HCl with a reaction time of 0, 1, 2, 3, 4, and 5 h for CNC-C0, CNC-C1, CNC-C2, CNC-C3, CNC-C4, and CNC-C5; and for a temperature of 100 °C, hydrolysis using 3 M HCl with a reaction time of 0, 1, 2, 3, 4, and 5 h for CNC-C0, CNC-C1, CNC-C2, CNC-C3, CNC-C4, and CNC-C5.

The hydrolysis process was carried out continuously with time intervals every 1, 2, 3, 4, and 5 h. Once every 1 h a sample was taken for analysis, and the remaining sample was continued for the next hour. After reaching the cellulosic suspension sampling time, sufficient distilled water was added to the suspension obtained to stop the acid reaction, the sample was decanted in a bottle for 1 to 2 days to precipitate particles of crystalline cellulose, and then it was washed with distilled water until neutral, centrifuged, and freeze-dried. The resulting CNCs were characterized.

CNC yield in each fraction was calculated by dividing the CNC weight in each fraction by the initial weight of cellulose, where:$$yield=\frac{CNC\;weight}{cellulose\;weight}\times 100\%$$

### 2.4. Characterization

#### 2.4.1. Chemical Structure Analysis

OPEFB, cellulose, and CNC chemical structures were characterized with a Nicolet 8700 Fourier transform infrared (FT-IR) spectrophotometer. FT-IR spectra were recorded in the spectral range of 4000–400 cm^−1^. Samples in the form of cellulose fiber and fine crystalline nanocellulose powder were placed on the KBr plate, palletized, and then placed in a sample container and analyzed at wave numbers of 400–4000 cm^−1^ to see the functional groups of lignin and cellulose.

#### 2.4.2. Morphological Analysis

The CNC surface morphology was analyzed using Oxford INCA 400 voltage 15 KV scanning electron microscopy (SEM) models. The nanocellulose crystal powder sample was placed in the sample holder, and then the sample was glued with a stab made of palladium specimen metal. Then, the sample was cleaned with a blower and coated with gold and palladium in a dispatcher machine with a pressure of 1492 × 10^−2^ atm. The sample was then put in a particular room and irradiated with a 10 kV electron beam so the sample would emit secondary electrons and bounced electrons, which could be detected by the sensor detector, which was then amplified with an electric circuit that created a CRT (cathode ray tube) image. Then, it was zoomed in on to capture a good photo.

#### 2.4.3. Crystal Structure Analysis (XRD)

The crystal structures were characterized on a Philips PZ1200 X-ray diffractometer (XRD) using Cu Kα X-rays with a voltage of 40 kV and a current of 30 mA. Samples of fine crystalline powder, as much as 0.5 mg, were coated and placed in a sample tool and then analyzed. X-ray diffraction data were collected over an angular range of 0–50 in steps of 0.02° at room temperature. The crystallinity of the samples was determined by the method of Segal:(1)$$Crystallinity=\frac{I(crystalline)-I(amorphous)}{I(crystalline)}\times 100\%$$
where *I*_200_ is the intensity of all the crystalline peaks at 2θ between 22° and 23°, and *I_am_* is the intensity of the diffractions in the same units at 2θ = 18° and 19°, which is a non-crystalline location [27].

The peak widths of the crystals were estimated from plane 2 0 0 (cellulose I), determined using the equation of Scherrer:(2)$$D_{\mathrm{hkl}}=\frac{K\cdot \lambda }{Bhkl\;\cdot cos\theta }$$
where *Dhkl* is the peak widths in the direction normal to the *hkl* family of lattice planes, *K* is the Scherrer constant (1.00 for equatorial reflections of rod-like or needle-like crystallites), *λ* is the wavelength of the radiation (1.54 Å), and *Bhkl* is the full width at half-maximum (FWHM) in radians of the reflection of that family of lattice planes [28].

#### 2.4.4. Thermogravimetric Analysis

The thermal stability was studied using a NETZSCH TG 209 F1 thermogravimetric analyzer. Samples in the form of fine crystalline powder were placed in the apparatus, and then the samples were heated from room temperature to 500 °C at a heating rate of 25 °C min^−1^ under a nitrogen atmosphere with a flow rate of 30 mL min^−1^.

#### 2.4.5. Particle Size Analysis by PSA

The particle size distribution and zeta potential of the CNC suspensions were measured with a Nano ZS Malvern Zetasizer, by second multi-angle particle size analysis based on dynamic light scattering, with a particle size analyzer (PSA), and via low-angle zeta potential analysis by electrophoretic light scattering (ELS). A 2.5 mg/mL CNC suspension was prepared, and measurements were made three times at 25 °C.

## 3. Results and Discussion

### 3.1. CNC Preparation Conditions

The effects of the preparation conditions on the CNC yield obtained by the hydrolysis process using hydrochloric acid under sonication–hydrothermal conditions are summarized in Table 1. The acid:cellulose ratio that was used for the hydrolysis reaction was 60 mL/g for (1; 2; 3; 4; 5) Molar HCl concentration runs and reaction times of 0 and 1 h, with a constant temperature of 110 °C. Previously, sonication was carried out for 30 min with a frequency of 28 KHz on all experimental variables. The role of sonication in this experiment was for swelling, which aims to cause the cellulose chains to become stress. Then, the amorphous part can easily degrade hydrothermally, and then the crystal chains are fractionated to become shorter and nano-sized.

The reaction time of 0 h was the process’s start time until it reached a constant temperature of 100 °C, 110 °C, and 120 °C. The suitable temperature for hydrolyzing cellulose into the CNC product was 110 °C; if the hydrolysis temperature was lower, it was assumed that the substrate was not entirely formed because it still had an amorphous part of the chain; otherwise, if the hydrolysis temperature was higher than 120 °C, it caused the CNCs to be degraded into glucose, with low yield and crystallinity [29]. The preparation conditions for the 3M acid catalyst involved hydrolysis with a temperature at 120 °C, and reaction times of 0, 1, 2, 3, 4, and 5 h showed an even lower yield and CNC crystallinity due to the quick crystallinity degrading into glucose and hydroxymethylfurfural (HMF). The hydrolysis with the temperature at 100 °C with the 3 M HCl catalyst and reaction times of 0, 1, 2, 3, 4, and 5 h showed a yield not yet fully formed, where the amorphous chain was not completely broken, and the fragments’ crystal segments had not been correctly formed.

The effect of hydrolysis temperature, as in Table 1, showed a significant difference in the yield and percentage of crystallinity obtained. If the temperature was low, the cellulose chains were not able to break off all the amorphous parts and the crystal size was still long. On the other hand, if the temperature was higher, it caused many cellulose crystal chains to be degraded and decompose into glucose, so the yield was low.

The reaction time and concentration of the experiment also was very influential on the product, where the experimental hydrolysis process with 1, 2, 3, 4, and 5 M hydrochloric acid at 1 h reaction time under a sonication–hydrothermal process at 110 °C showed a high yield and crystallinity, with a yield of 52.2, 79.09, 74.8, 72.9, and 60.5%, respectively. The reaction time was shorter because the hydrolysis process was assisted by ultrasonic vibrations and hydrothermal pressure, which can cause the cellulose chains to break more easily into shorter sizes. 

Compared with research by Hastuti, OPEFB-based cellulose dissolved in 3 M HCl had a reaction time of 3 h to produce CNCs, without an ultrasonic bath or hydrothermal pressure to assist the hydrolysis process, and the yield obtained was very low. The yields of the CNC-A, B, and C from the three pulps, i.e., pulp-A, -B, and -C, were 21, 18, and 19%, respectively. Other different conditions for acid concentrations, temperatures, and reaction times resulted in lower yields of CNCs with unclear birefringence in this work [30]. The crystallinity indices of the resultant CNCs by Hastuti also were lower, namely, CNC-A, B, and C were 65, 60, and 53%, respectively [30].

The 3 M HCl concentration became the point to determine the yield and crystallinity at hydrolysis temperatures of 100 °C and 120 °C with reaction times of 1, 2, 3, 4, and 5 h, because the concentration of 3 M HCl at a temperature of 110 °C had the highest yield and crystallinity. Furthermore, CNC-A1, CNC-B1, CNC-C1, CNC-D1, and CNC-E1 which were hydrolyzed at 110 °C were analyzed of FT-IR, SEM, PSA, and TGA-DTG, because they has crystallinity and yield percentage of highest. underwent further processes, namely, FT-IR, SEM, PSA, and TGA-DTG, because they had the highest crystallinity and yield percentage.

Hydrolysis using 4 M and 5 M HCl at 110 °C showed that the suspension color would be dark. High acid concentration can result in excessive degradation of cellulose, in which acid can penetrate quickly into the network layer and hydrolyze amorphous regions that subsequently hydrolyze crystalline regions of cellulose—decreasing yield also allegedly due to prolonged reaction times, which can degrade cellulose crystalline regions and cause carbonization, so the color will be dark. According to Yu, the hydrolysis reaction was limited by a relatively short reaction time, so the acid decreased only in the amorphous regions of the cellulose and left the crystals. Thus, the reaction time was one of the most critical parameters to increase the yield [23].

Figure 1 shows the effects of concentration on the CNC yield at 1 h of reaction time under hydrolysis by sonication–hydrothermal conditions at a temperature of 110 °C. The graph shows the yield increased when using 2 M HCl, then the yield decreased when using 3, 4, and 5 M HCl. The reaction time was 1 h, the most stable conditions to produce the desired material. Sonication can accelerate the swelling process of the cellulose chains so that they are easily broken down into crystal fractions during the hydrothermal method and then form nanocrystal chains with shorter sizes. So, combining the sonication–hydrothermal process can accelerate the process of creating nanocellulose-sized crystals and maintain high crystallinity without being degraded much into glucose and HMF. It can be seen in Table 1 that in just 1–2 h, the hydrolysis process formed high CNC crystallinity and maintained high yields as well.

### 3.2. Scission Mechanism of Cellulose Chains into CNCs

Sonication–hydrothermal hydrolysis by HCl is a process that speeds up the decomposition of amorphous parts, breaks the crystal chain into short-sized segments/parts, and reduces the degradation of cellulose crystals, resulting in high yield and crystallinity. The mechanism of cellulose hydrolysis was carried out by initial hydrolysis to remove amorphous regions from the cellulose polymer, then fragmentation of crystal segments to produce nanocrystals, as shown in Figure 2. The amorphous areas in the cellulose were easier to decompose using hydrochloric acid than crystalline areas so that crystals could be created.

### 3.3. FT-IR Analysis

FT-IR analysis was carried out to see the difference in the absorption peak of the functional group through a specific wave number, which indicates a difference between the OPEFB, cellulose, and CNC spectrum. The typical FT-IR spectrum of OPEFBs and cellulose shown in Figure 3 and the placement of the vibration area are summarized in Table 2. Figure 4 shows the FT-IR typical spectra from (a) CNC-A1 (1 M, 1 h), (b) CNC-B1 (2 M, 1 h), (c) CNC-C1 (3 M, 1 h), (d) CNC-D1 (4 M, 1 h), and (e) CNC-E1 (5 M, 1 h), with a hydrolysis temperature of 110 °C. All the samples showed two main absorption areas, as shown by Haafiz, namely, in the areas of high wave number (2800–3500 cm^−1^) and lower wave number (500–1700 cm^−1^) [31]. The spectrum in Figure 4 shows broad absorption peaks located at 3250–3500 cm^−1^, which was an –OH group stretching. Absorption peaks at the 2897–2917 cm^−1^ region are associated with the–CH_2_ group. The absorption peak of 2897 cm^−1^ was overlapping with the –CH_2_ ribbon. This peak was only in OPEFBs and cellulose raw materials, while the CNC peak was lost as a result of the termination of amorphous cellulose chain. Jahan reported the same results [32].

Figure 3 shows the absorption in the 1600–1650 cm^−1^ for the OPEFB and cellulose samples, indicating the absorption of water or–OH stretching. According to Johar, the peak is related to the bending form of water molecules due to a strong interaction between the cellulose and water [33]. In contrast, the absorption peak of the sample was gone. The structure became crystalline cellulose, no longer binding with water. The absorption peaks at 1454 cm^−1^ and 1419 cm^−1^ were the suspected area bending vibration of O-C-H derived from lignin components; it was consistent with that obtained by Yu [23]. The second peak only appeared in the OPEFB spectrum, while the CNCs and cellulose spectrums did not exist. This indicates that the lignin removal took place ideally. In addition, the absorption peaks in the region of 1300–1365 cm^−1^ in all samples in the vibration band of C-H and C=O were associated with an aromatic ring polysaccharide, which is consistent with the absorption band analyzed by Nacos [34].

Absorption peaks at 1238 cm^−1^ in Figure 4 were caused by the C-O-C of aryl–alkyl contained in lignin; this peak was not shown in the CNC or cellulose spectrums. The absorption peaks at 1160 cm^−1^, 1158 cm^−1^, and 1104 cm^−1^ shown in the CNC and cellulose spectrums were caused by the deformation of the vibration of C-H and C-O-C pyranose; this is consistent with the results demonstrated by Kargarzadeh [35]. Changes in the spectral characteristics of cellulose demonstrated the elimination of hemicellulose and lignin. All CNCs in Figure 3 showed an increase in the intensity of the band at 1053 cm^−1^, which depicted the stretch ring pyranose C-O-C. According to Corrêa, this implies there was an increase in the value of crystalline cellulose [36]. These results are consistent with the results obtained from X-RD, where the CNC substrate showed the growing crystals. Absorption peaks at 896 cm^−1^ were the lowest C-H vibration of cellulose (anomeric vibrations, specific for β-glucosides) [37].

### 3.4. Morphological Analysis

The CNCs and cellulose morphologies were investigated using SEM, as shown in Figure 5. Micrographs show the difference between the cellulose and CNC materials hydrolyzed with 1, 2, 3, 4, and 5 M HCl under a sonication–hydrothermal process at 110 °C, as shown in Figure 5. The reaction time and concentration of hydrochloric acid yielded decisive results and size distribution. The surface structure of CNC-C1 and CNC-D1 had a smaller dimension of 2 μm in width, from 1.0 to 9.0 mm and from 1.0 to 9.5 mm, while the CNC-B1 had a width dimension of 10 μm and 10.5 mm particles. This shows the effect of acid concentration on the size of the particle fineness. The fineness of the crystal greatly influences the morphological changes, so the fineness of the crystals needs to be considered to provide a better surface structure. According to Haafiz, acid hydrolysis can make the fiber morphology of MCC softer than OPEFB [31].

Overall, CNCs showed a more regular surface structure than cellulose, which was more random. This indicates that cellulose crystalline regions can withstand attacks of hydrochloric acid, and hydrochloric acid can remove the amorphous component of cellulose, resulting in better morphology. The sonication–hydrothermal process shown in the following CNC morphology can transform the chain of native cellulose into crystalline cellulose with a shorter dimension than the cellulose morphology.

The AFM images in Figure 6 demonstrate the CNC-C1a width and length; acidic hydrolysis could reduce the size of the cellulose due to the elimination of an amorphous chain and simultaneously reduce the size to form cellulose nanocrystals. CNC-C1a showed the size of the crystals had a diameter of about 42 nm and a length of 268 nm. The CNC shape was typically a rigid, rod-shaped, mono-crystalline cellulose domain of about 16 nm in diameter and 230 nm in size [23]. This proves that the 3 M hydrochloric acid hydrolysis can maintain a cellulose crystal region, as evidenced by the still-high yield; the dimensions of the crystals were tiny, and almost all uniform. When viewing the SEM and XRD analysis results, the CNC sample showed similar results, with the best results for the CNC-C1 sample, so the AFM analysis sufficiently represented and referenced the CNC-C1.

Figure 6 for CNC-C1b shows the CNC morphology on a smaller scale, namely, 200 nm with a magnification of 30,000 times, and it had various particle diameters, including 332 nm and 118 nm. The level of crystal fineness strongly influences CNC morphology, so it was necessary to pay attention to the fineness of the crystals to provide a smaller and better surface structure. The particle size distribution was not uniform due to strong agglomeration between individual particles, making it difficult to see the morphology of a single particle. This agglomeration could have been caused by crystals that agglomerated during the drying process. To be more specific, the particle distribution and individual crystal sizes can be seen by using a particle size analyzer (PSA), as shown in Figure 7.

### 3.5. X-ray Diffraction Analysis

The CNC XRD patterns prepared through hydrolyzing with 1, 2, 3, 4, and 5 M HCl with a reaction time of 1 h under the sonication–hydrothermal process are presented in Figure 7. The crystallinity and crystallite size are shown in Table 1.

All diffraction patterns from cellulose crystals shown in Figure 7 were indicated by peaks around 2θ = 16°, 22,6°, and 35° and minimum diffraction pattern for the amorphous field around 2θ = 18°. According to Yu and Rosli, this is the hallmark of the structure of cellulose crystals [23,38].

Cellulose obtained from OPEFBs had a crystallinity of 63.02%; this value is high compared to the resulting nanofibers hydrolyzed from OPEFBs using H_2_SO_4_, which was 59% [39]. The high crystallinity obtained due to the elimination of hemicellulose and lignin in the amorphous area led to the arrangement of the cellulose molecules [37]. Compared to cellulose, CNCs have a broader peak because chloride acid is more efficient in removing non-cellulosic polysaccharides and causing the dissolution of the amorphous zone. The crystallinity of the resultant CNCs by Hastuti also was lower compared to in our research; namely, CNC-A, B, and C were 65, 60, and 53%, respectively [30]. Meanwhile, in our research, CNC-A1, CNC-B1, and CNC-C1 were 70.63, 75.87, and 78.59%, respectively.

The CNC-A1 crystallinity obtained through 1 M HCl hydrolysis with a reaction time of 1 h was lower compared to CNC-B1 and CNC-C1, which were hydrolyzed using 2 M and 3 M HCl and reached 75.87% and 78.59%. This was due to 3 M HCl being able to attack the amorphous region of cellulose and degrade completely to leave a crystalline part. Still, at the same time, the yield tends to decrease because the attack of hydrochloric acid also degrades the crystal portion. According to Li and Spagnol, the high crystallinity is due to the removal of amorphous regions of cellulose, which encourages hydrolytic cleavage of glycosidic bonds, ultimately releasing individual crystals [37,40]. To defend the crystallinity and high yield, the acid concentration must be considered, and the hydrolysis period must be shorter, as shown in Table 1, where the longer the reaction time, the lower the yield obtained. The same thing was also seen when the acid concentration was increased to 4 M and 5 M, showing CNC-D1 and CNC-E1, indicating that the yield tended to decrease sharply and even the crystallinity decreased. Using an increasingly concentrated hydrochloric acid and reaction time was long believed to be the cause of the degraded cellulose crystal region so that yield and crystallinity decrease the darker the color of the crystals; the same thing was also conveyed by [41].

The conditions of the hydrolysis process affect the resulting crystallinity, where a higher acid concentration used causes the crystallinity to decrease and the yield to be lower, because the higher the acid concentration, the easier it is to attack the crystal area and cause the crystals to degrade in a shorter time, thus causing the resulting yield to be lower. This can be seen in the XRd diffractogram pattern images of CNC-D1 and CNC-E1. A dilemma in CNC preparation is when you want to increase the crystallinity but will decrease the yield; on the other hand, if the result is increased, there is a tendency that not all of them will become crystals, and there will still be many amorphous parts. According to Yu, an acid treatment for a long time can weaken the cellulose crystalline regions, even resulting in carbonization [23]. Based on kinetic tests carried out by Zulnazri, the reaction time greatly influences the crystallinity of CNC, where based on the hydrolysis results, a crystallinity of 78.59% was obtained with a reaction time of 1 h, capable of removing the amorphous part of the cellulose chain, leaving the cellulose crystalline part, when the time was up. the reaction was increased to 2 h causing the crystallinity to drop to 77.07% and the 1 h time based on reaction kinetics already produces constant crystallinity so that the subsequent degradation process to produce glucose can also be less [42]. 

All the X-ray diffraction pattern showed a broad diffraction peak at 2θ = 22.6° with 002. The peak at about 34.5° was a group of crystalline peaks that included the 004 reflection. The spacing (plane 004) is related to the crystallographic unit of cellulose and was relatively shifted along the molecular axis between the two chains. All XRD patterns in Figure 7 show the presence of a weak crystal peak in the 004 plane, which indicates that hydrolysis with hydrochloric acid was able to provide relatively stable conditions to maintain the crystal chains. According French, in the XRD pattern it can be seen that the non-hydrolyzed materials have a predominance of crystalline domains typical of cellulose I, as evidenced by the presence of peaks at 2 = 15° (plane 1–1 0), 22.5° (plane 2 0 0) and 34° (plane 0 0 4) [43].

The diffraction data also showed high CNC crystallinity and cellulose I without cellulose II, which is indicated by the absence of doublet at 22.6° [17,44]. The peak diffraction located at 22.6° became sharper, showing an increase in crystallinity, as shown by Haafiz [31]. Increased crystallinity is related to the increased rigidity of the cellulose structure to produce high tensile strength [17,31,45].

To find out the cellulose crystal size, D_hkl_ was calculated using Equation (2). Dhkl is an important parameter to determine whether the crystals obtained were nano- or micro-sized. Table 1 shows that the crystal size tended to be more minor with the high concentration of acid used, which caused hydrochloric acid to attack and fragment the crystal segments more quickly, so the crystal size was smaller. But there were differences in CNC-B0 and CNC-B1, which showed smaller crystal sizes. Compared to CNC-C1, this condition was suspected of having a powerful molecular collision in the reactor during hydrolysis. The crystal size also affects the type of acid used, where hydrolysis with sulfuric acid tends to provide a smaller crystal size. Still, more crystals were degraded, and the yield obtained was small, as Yu and Bondeson stated [23,46]. CNC-D0 showed the smallest crystal size of 2.61 nm (26.1 Å) at 110 °C, and this suggests that the hydrolysis process under the applied sonication–hydrothermal conditions was very effective in reducing the crystallite size of cellulose.

**Figure 7 polymers-16-01866-f007:**
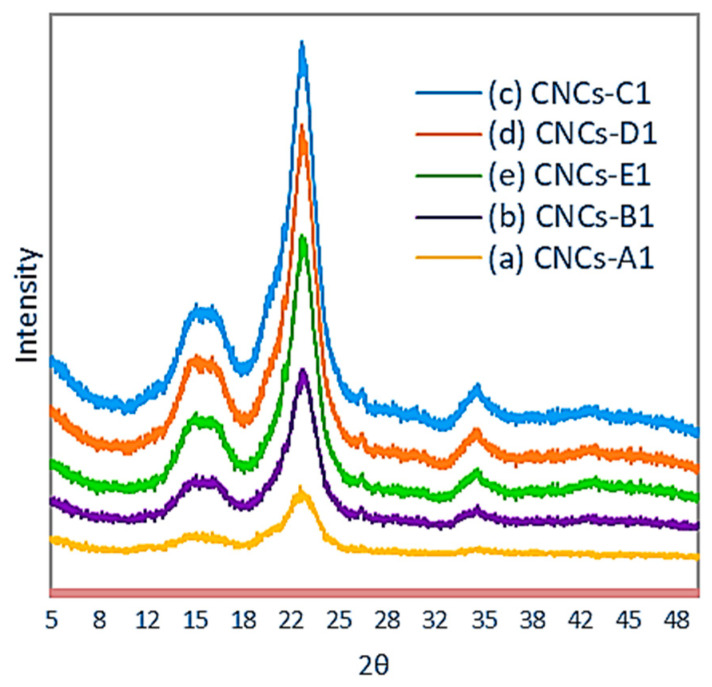
X-ray diffractograms pattern of (a) CNC-A1 (1 M, 1 h), (b) CNC-B1 (2 M, 1 h), (c) CNC-C1 (3 M, 1 h), (d) CNC-D1 (4 M, 1 h), and (e) CNC-E1 (5 M, 1 h), obtained with a hydrolysis temperature of 110 °C.

### 3.6. Thermogravimetric Analysis

The thermal stability was studied using a NETZSCH TG 209 F1 thermogravimetric analyzer. The samples were heated from room temperature to 500 °C at a 25 °C/min-1 heating rate under a nitrogen atmosphere with a flow rate of 30 mL min^−1^. Figure 8 shows the curve of the thermogravimetric analysis (TGA), and Figure 9 shows the curve of the derivative thermogram (DTG), whereby samples were prepared through different route preparations under sonication–hydrothermal conditions and degradation onset temperature (T_0_), and the maximum decomposition temperature (T_max_) is listed in Table 3.

Thermal degradation of the product involves depolymerization, dehydration, and decomposition of the glycosyl unit, followed by the formation of charred residue [23]. The curve shows that the degradation temperature (TGA) and decomposition temperature (DTG) were getting more prominent and sharper down, which indicates that the glycosyl unit was depolymerized and decomposed until the formation of the charred residue required a high calorific value, which suggests that the CNCs were very crystalline and stiff.

The phenomenon of degradation temperature (T_0_) of CNC-A1 began at a lower temperature (294.81 °C) and decomposed at a temperature (T_max_) of 326.72 °C; and the T_0_ of CNC-B1 began at a lower temperature (305.20 °C) and decomposed at a temperature (T_max_) of 340.15 °C; the level of thermal stability of CNC-C1 was almost the same as that of CNC-B1, which could be maintained until a temperature (T_0_) of 305.66 °C and then decomposed at the maximum temperature (T_max_) of 339.82 °C. CNC-D1 began degrading at temperatures of 307.09 °C and decomposed at 340.56 °C. A higher TGA and DTG indicate more crystallinity, as seen in CNC-B1, CNC-C1, and CNC-D1. This is relevant to the degree of crystal on XRD, where CNC-C1 and CNC-D1 had prominent crystallinity.

Overall, the three CNC samples had almost the same thermal stability. This proves that cellulose hydrolysis with hydrochloric acid under sonication–hydrothermal conditions can maintain thermal stability, so the degradation temperature and high-temperature decomposition were obtained.

The T_max_ in Hastuti’s research showed good thermal stability for CNC-A, CNC-B, and CNC-C at 358.5, 359.2, and 346.5 °C, respectively [30]. Hastuti also reported that EFB had higher thermal stability, namely, 347–359 °C as the maximum degradation temperature, compared to wood CNC made through sulfuric acid hydrolysis (15 and 311 °C, respectively) [30]. The use of hydrochloric acid and ultrasonication for EFB hydrolysis is effective for producing crystalline CNCs with long-term nano-dispersibility. Moreover, according to Corrêa, hydrolysis with hydrochloric acid can form crystals with a relatively small size distribution, resulting in a single peak in the DTG curve [36].

### 3.7. The CNC Size

Figure 10 shows the CNC size hydrolyzed using 1, 2, 3, 4, and 5 M HCl with a reaction time of 1 h at 110 °C. The particle size analyzer aimed to see the distribution of particles and the average particle size in cellulose nanocrystals. The analysis results showed that the particle size was tiny, below 100 nm. This indicates that these particles were characteristic of nanocrystals below 100 nm. This condition also shows that the sonication and hydrothermal processes could reduce the particle size and fragment the crystal segments into nanoscale; they also degraded the amorphous chains into simple glucose and sugar but left the crystal segments. From Table 4, it can be seen that CNC-A1 had particles with a diameter of 65.93–303.50 nm, CNC-B1 has a particle diameter of around 6.25 nm, CNC-C1 had particles with a diameter of 5.83 nm, and CNC-D1 had a particle diameter of 1.013 nm.

In comparison, CNC-E1 has a particle diameter of 4.64 nm. In general, it can be seen that the higher the acid concentration used, the smaller the CNC diameter; this proves that the acid concentration was significant in reducing the particle size, but for CNC-E1, which was hydrolyzed with 5 M HCl, the particle diameter was slightly larger than CNC-D1, which was very small. This may have been caused by uncontrolled process conditions. The higher the acid used, the easier it was to hydrolyze the cellulose chains, where the acids quickly penetrated the inner layers of the cellulose chains and broke the crystal chains into smaller individuals. These data support the SEM analysis, where the smaller the particle size, the better the surface structure.

### 3.8. CNC and Cellulose Comparisons

Cellulose extracted from OPEFBs appeared coarser and had irregular fibers compared to nanocellulose crystals, as shown in Figure 11. After sonication–hydrothermal hydrolysis of cellulose, a suspended liquid was obtained, which was still mixed with acid solvents. Some parts of the suspension still appeared as coarse-sized particles and quickly settled to the bottom of the bottle as residue. At the same time, the nanocrystal-sized portion was almost invisible to the eye because it was excellent in size, suspended in a liquid that floated on top of the residue. To obtain the exemplary nanocrystals, the suspension was passed through the dialysis membrane for 2 × 24 h so nano-sized particles with a smooth and uniform size could be taken. The cellulose crystals obtained were in the form of fine particles and white crystals. The results of the SEM and PSA tests showed that the particles were under 100 nm in size and uniform in shape. Meanwhile, the results of the analysis of crystallinity with XRD showed that there were crystals at 2θ = 22.6° with 200 regions and had a high crystallinity index of 78.59%. These results indicate that nanocellulosic crystals can be used as a raw biomedicine material.

## 4. Conclusions

A study on the manufacture of CNC has been presented through hydrochloric acid hydrolysis of cellulose-based OPEFB raw materials under sonication–hydrothermal conditions. The chemical structure of cellulose and CNCs showed a deletion of lignin and hemicellulose shown on FT-IR spectra. CNCs showed that the morphology was more compact and had a smaller fragment compared to cellulose. XRD analysis showed that the CNCs had a high crystallinity index. It was proven that hydrochloric acid hydrolysis could preserve the cellulose I crystal structure at around 2θ = 22.6°. High CNC crystallinity (78.59%) could be achieved by hydrolysis using 3 M HCl. TGA analysis showed that the CNCs obtained from OPEFB biomass had good thermal stability. The hydrothermal process at 4 M HCl concentration and 1 h reaction time showed TGA and DTG values of 307.09 °C and 340.56 °C.

## Figures and Tables

**Figure 1 polymers-16-01866-f001:**
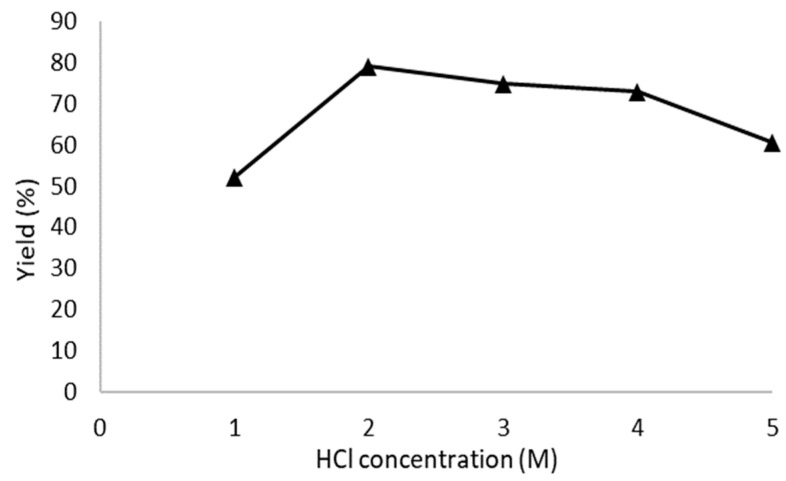
Effect of acid concentration on the yield of CNC-A1, CNC-B1, CNC-C1, CNC-D1, and CNC-E1 with a reaction time of 1 h, hydrolyzed at 110 °C.

**Figure 2 polymers-16-01866-f002:**
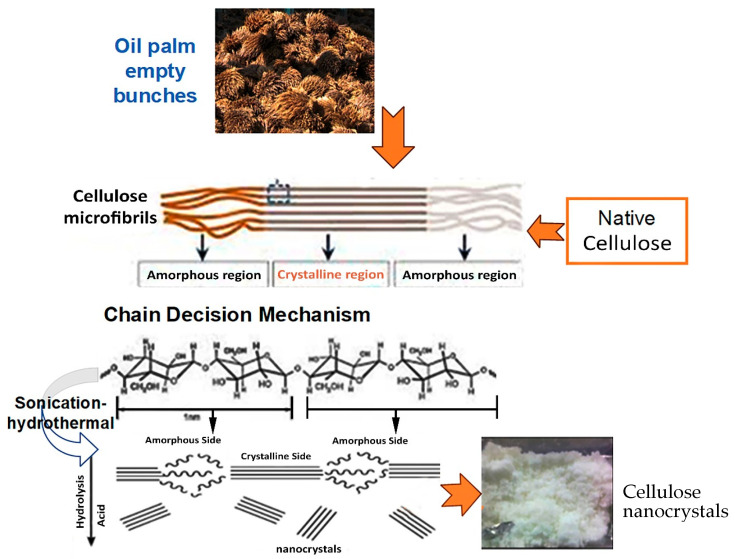
Chain scission mechanism.

**Figure 3 polymers-16-01866-f003:**
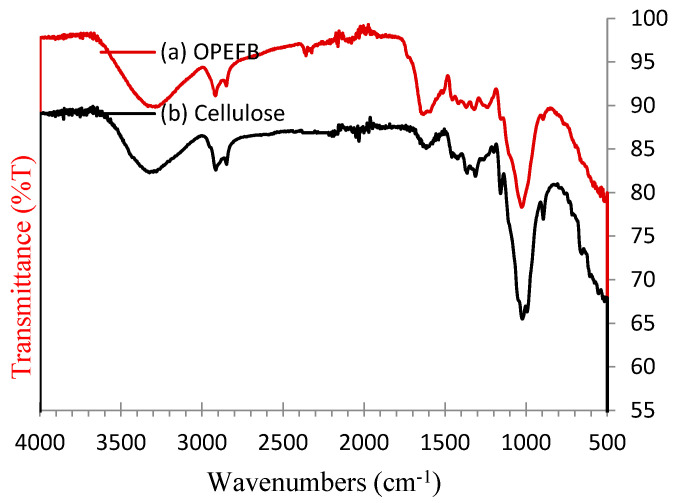
FT-IR spectra obtained from (a) OPEFB pulp and (b) cellulose.

**Figure 4 polymers-16-01866-f004:**
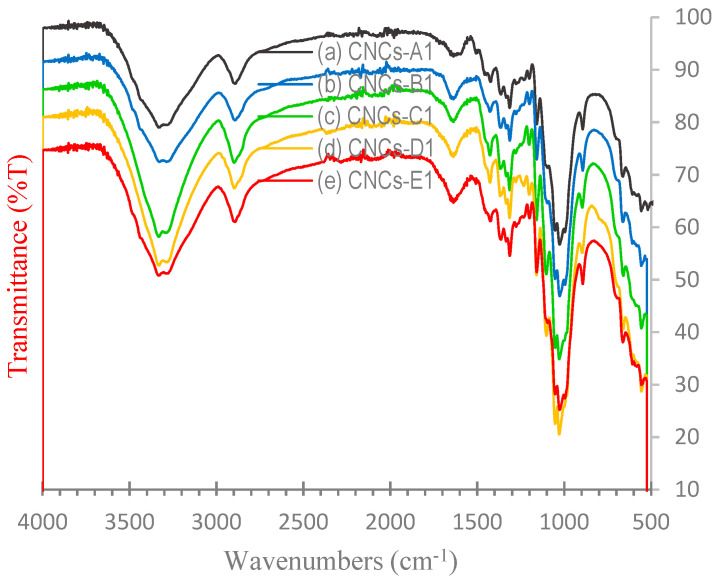
FT-IR spectra obtained from (a) CNC-A1 (1 M, 1 h), (b) CNC-B1 (2 M, 1 h), (c) CNC-C1 (3 M, 1 h), (d) CNC-D1 (4 M, 1 h), and (e) CNC-E1 (5 M, 1 h), with a hydrolysis temperature of 110 °C.

**Figure 5 polymers-16-01866-f005:**
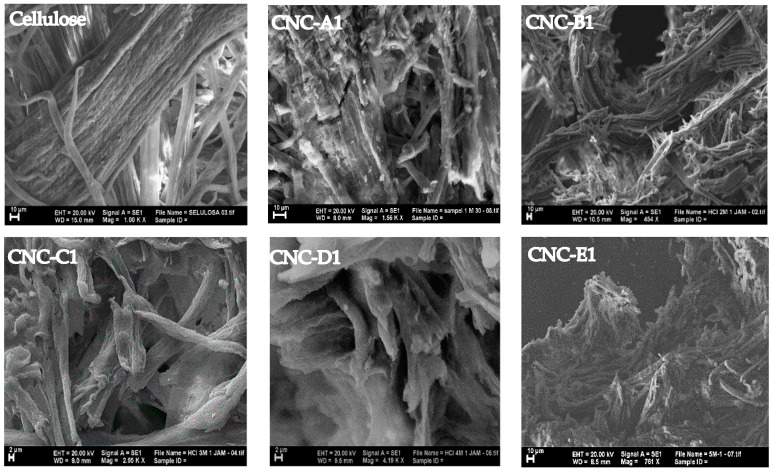
Typical scanning electron micrographs of cellulose, CNC-A1 (1 M, 1 h), CNC-B1 (2 M, 1 h), CNC-C1 (3 M, 1 h), CNC-D1 (4 M, 1 h), and CNC-E1 (5 M, 1 h).

**Figure 6 polymers-16-01866-f006:**
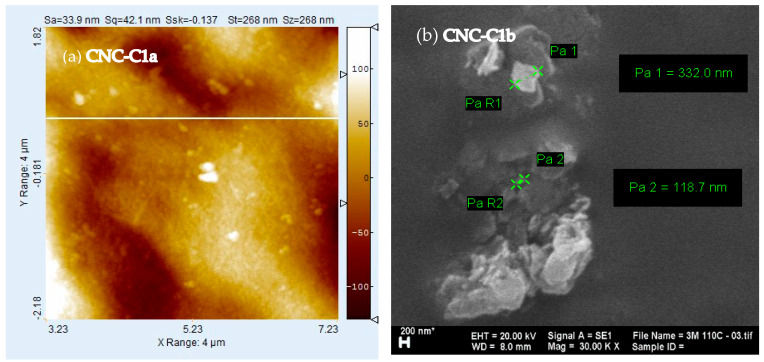
Typical atomic force microscopy of CNC-C1a and 200 nm-scale micrograph SEM of CNC-C1b by the sonication–hydrothermal process at 110 °C.

**Figure 8 polymers-16-01866-f008:**
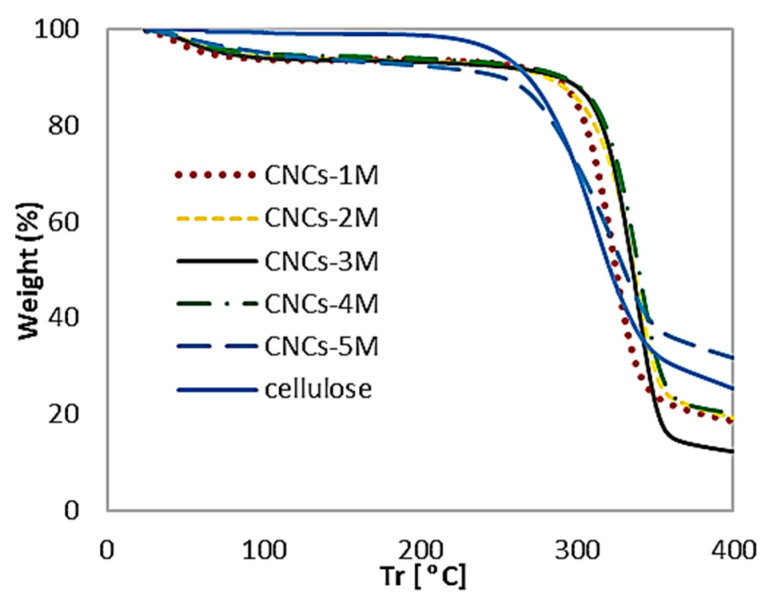
TGA thermogram of cellulose, CNC-A1 (1 M, 1 h), CNC-B1 (2 M, 1 h), CNC-C1 (3 M, 1 h), CNC-D1 (4 M, 1 h), and CNC-E1 (5 M, 1 h), obtained with a hydrolysis temperature of 110 °C.

**Figure 9 polymers-16-01866-f009:**
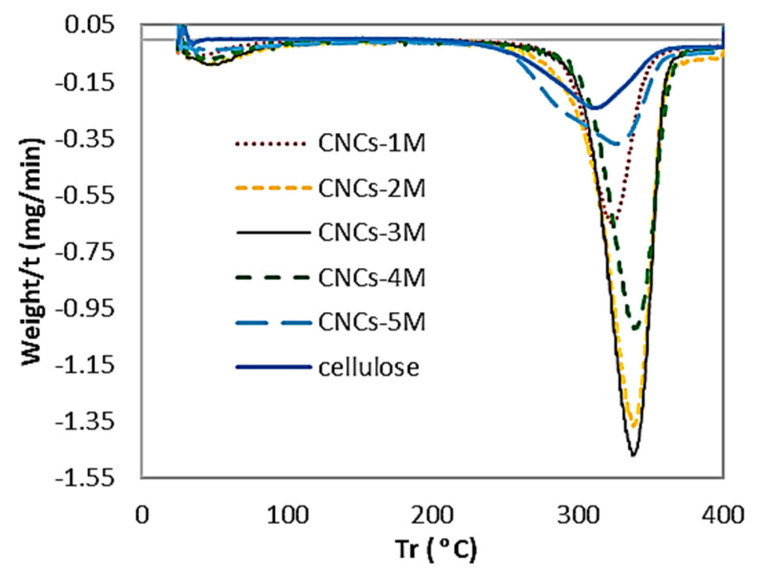
DTG thermogram of cellulose, CNC-A1 (1 M, 1 h), CNC-B1 (2 M, 1 h), CNC-C1 (3 M, 1 h), CNC-D1 (4 M, 1 h), and CNC-E1 (5 M, 1 h), obtained with a hydrolysis temperature of 110 °C.

**Figure 10 polymers-16-01866-f010:**
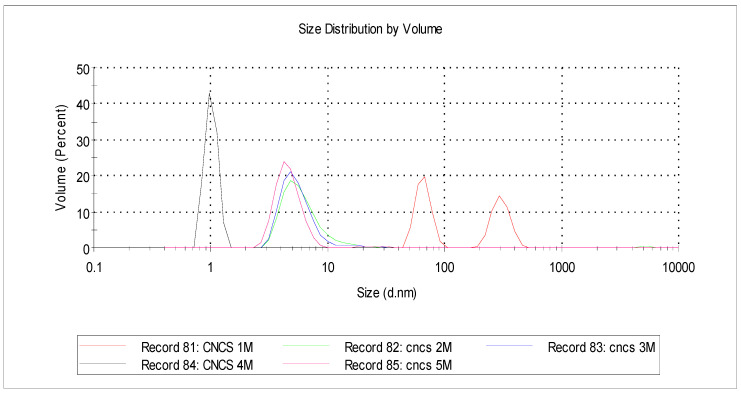
Size distribution obtained by particle size analyzer of CNC-A1 (1 M), CNC-B1 (2 M), CNC-C1 (3 M), CNC-D1 (4 M), and CNC-E1 (5 M) at 1 h reaction time under hydrothermal conditions at 110 °C.

**Figure 11 polymers-16-01866-f011:**
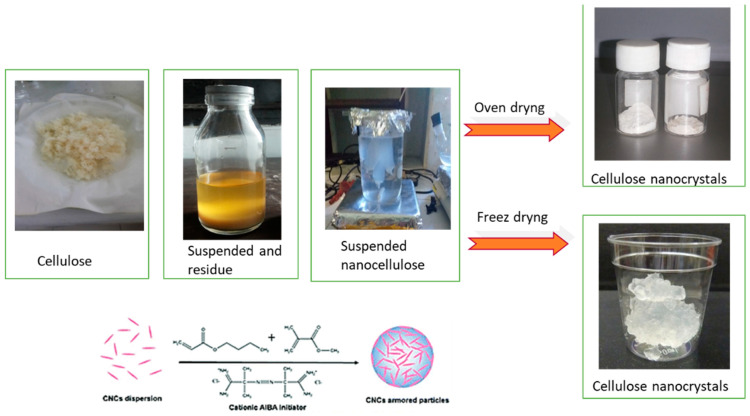
Cellulose and cellulose nanocrystals.

**Table 1 polymers-16-01866-t001:** Yield of CNCs based on preparation conditions, T 110, 120, and 100 °C.

Sample	Hydrolysis HCl (M)	Reaction Time (h)	Yield (%)	Crystallinity (%)	*D_hkl_* _(002)_ (nm)	PSA (nm)	(TGA) T_0_(°C)	(DTG) T_m_(°C)	Temp. Process (°C)
CNC-A0	1	0	35.20	65.32	15.64	-	-	-	
CNC-A1	1	1	52.20	70.63	15.64	65.93	294.81	326.72	
CNC-B0	2	0	66.80	72.82	4.69	-	-	-	
CNC-B1	2	1	79.09	75.87	4.69	6.25	305.20	340.15	
CNC-C0-110	3	0	48.90	73.40	11.74	-	-	-	110
CNC-C1-110	3	1	74.82	78.59	7.83	5.83	305.66	339.82	
CNC-D0	4	0	58.50	67.75	2.61	-	-	-	
CNC-D1	4	1	72.90	77.19	3.35	1.013	307.09	340.56	
CNC-E0	5	0	45.82	68.77	4.69	-	-	-	
CNC-E1	5	1	60.54	76.36	2.93	4.64	269.53	337.87	-
CNC-C0-120	3	0	45.64	57.70	3.91	-	-	-	
CNC-C1-120	3	1	63.70	61.22	2.35	-	-	-	
CNC-C2	3	2	54.05	64.05	3.88	-	-	-	120
CNC-C3	3	3	40.50	58.91	17.46	-	-	-	
CNC-C4	3	4	31.02	57.89	17.46	-	-	-	
CNC-C5	3	5	25.07	57.17	3.79	-	-	-	
CNC-C0-100	3	0	48.10	64.16	3.88	-	-	-	
CNC-C1-100	3	1	75.20	68.24	4.69	-	-	-	
CNC-C2	3	2	56.50	73.07	3.91	-	-	-	
CNC-C3	3	3	44.70	68.04	3.96	-	-	-	100
CNC-C4	3	4	48.00	66.02	3.35	-	-	-	
CNC-C5	3	5	24.32	64.75	3.35	-	-	-	
Cellulose	NaOH extracted	2	50.04	63.02	22.9	-	233.43	314.17	

**Table 2 polymers-16-01866-t002:** FT-IR spectral peak assignments for OPEFB pulp, cellulose, and CNCs.

Peak Frequency (cm^−1^) for OPEFB Pulp, Cellulose, and CNCs	Peak Assignment
3250–3500	O-H bending
2917, 2914, 2897	CH_2_ groups
1600–1650	O-H stretching
1300–1450	CH_2_ aromatics
1238	C-O-C aryl–alkyl
1104, 1158, 1160	C-O-C stretching

**Table 3 polymers-16-01866-t003:** Thermal stability of CNCs.

Sample	Thermograv. Analysis (TGA) T_0_ (°C)	Derivative Thermograv (DTG) T_max_ (°C)
Cellulose	233.43	314.17
CNC-A1	294.81	326.72
CNC-B1	305.20	340.15
CNC-C1	305.66	339.82
CNC-D1	307.09	340.56
CNC-E1	269.53	337.87

**Table 4 polymers-16-01866-t004:** Particle size of CNCs hydrolyzed at a temperature of 110 °C.

CNC-A1		CNC-B1		CNC-C1		CNC-D1		CNC-E1	
% Volume	Size (d.nm)	% Volume	Size (d.nm)	% Volume	% Volume	Size (d.nm)	% Volume	Size (d.nm)	% Volume
54.7	6.25	99.3	5.83	99.5	54.7	6.25	99.3	5.83	99.5
45.3	357.90	0.1	631.8	0.1	45.3	357.90	0.1	631.8	0.1
	5014	0.6	4810	0.4		5014	0.6	4810	0.4
100		100		100	100		100		100

## Data Availability

Data are available within the article. Raw data are available upon reasonable request and express third-party consent.

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
