# Peer review of "Effect of Hydrochloric Acid Hydrolysis under Sonication and Hydrothermal Process to Produce Cellulose Nanocrystals from Oil Palm Empty Fruit Bunch (OPEFB)"

_polymers, 2024, doi:10.3390/polym16131866_

Round 1

Reviewer 1 Report

Comments and Suggestions for Authors

The manuscript reports the production of CNC via the acid-catalyzed sonication-hydrothermal hydrolysis of OPEFB. The topic of this study is interesting but there are some points that need to be improved.

1.    The authors mention the XRD in the experimental section, but the results and discussion only list the data on crystallite sizes. The XRD curves for all the samples with different synthetic conditions must be systematically provided in the manuscript or supporting information.

2.    FTIR spectra of the different hydrolysis conditions are suggested to be provided to observe the effect of the experimental conditions on the structures of the hydrolyzed products.

3.    There was no C-H stretching assignment in Table 2.

4.    The word “aril” in Table 2 is incorrect.

5.    Please also provide TEM images to support SEM images further.

6.    In order to recognize the sample code easily, the authors must recode it to provide some indication of the experimental conditions instead of A, B, C, etc. The description of the sample codes needs to be placed in the experimental section.

7.    At lines 145, 147, 327, and 329, the “Crl” abbreviation is unnecessary.

8.    What is the size of “5014” in the column of CNCs-A1 (Table 2)?

9.    There is a duplication of the %volume heading for the column of CNCs-C1 (Table 2).

10. Please provide the new Figures 4 and 7 with a higher resolution.

11. For all of the figures, please correct them to be consistent in using fonts and tick style. The recent ones are that some figures have ticks on axes, but some have not.

12. References are not updated. Please provide some more recent references.

13. There is no comparison of the CNCs prepared from this work with the ones reported in the literature. Please provide a comprehensive comparison of the properties of the CNCs prepared in this work and the literature to see the benefits of this current procedure over the other processes.

14. Section 3.9 is not the work that was carried out in this work. In order to provide perspectives, please do so briefly in the conclusion section. However, it would be better if the authors could conduct some more experiments for the application of the prepared CNCs.

15. The conclusion is needed to be revised to highlight the significant findings of this work. The recent one just lists the data.

16. There are numerous typos and formatting errors throughout the manuscript. Extensive English editing is required. 

Comments on the Quality of English Language

Extensive editing of English language required

Author Response

  1. In table 1, data on the degree of crystallinity (%) and crystal engraving (Dhkl) are included.
    The XRD curve is displayed on page 12 of Figure 5, complete with information and explanations in sub-chapter 3.5 and in detail the XRD values are available in Table 1. In one picture there are 5 curves with different process conditions including the XRD pattern curve of (a ) CNCs-A1 (1M, 1 hour), (b) CNCs-B1 (2M, 1 hour), (c) CNCs-C1 (3M, 1 hour), (d) CNCs-D1 (4M, 1 hour), and (e) CNCs-E1 (5M, 1 hour), for each name the curve is based on process conditions and can be seen in table 1, for example CNCs-A1 is a CNC crystal obtained by a hydrolysis process using 1 M HCl and a reaction time of 1 hour at a temperature of 110 oC
  2. FT-IR analysis with different conditions is available on page 8, Figure 3. namely: Typical FT-IR spectra obtained from 3a. (a) OPEFB-pulp, and (b) Cellulose, 3b. (a) CNCs-A1 (1M, 1 hour), (b) CNCs-B1 (2M, 1 hour), (c) CNCs-C1 (3M, 1 hour), (d) CNCs-D1 (4M, 1 hour ), and (e) CNCs-E1 (5M, 1 hour), with hydrolysis temperature 110°C. permission to explain that in the image 6 FT-IR spectra have been combined, and information is available on each spectrum, for example (a) CNCs-A1 is an image of a CNC spectrum obtained by the hydrolysis process using 1 M HCl and a reaction time of 1 hour with a temperature of 110 °C
  3. There is an explanation in sub-chapter 3.3 in paragraph 2 of the last line, namely: the absorption peak in the region 1300-1365 cm-1 in all samples is a vibration band from the C-H and C=O groups of cellulose which are associated with the aromatic ring of the polysaccharide. We can add this to the table. Thank you for the advice
  4. We have replaced the aryl in table 2 with aryl-alkyl.
  5. We are sorry that we did not test the surface structure study using TEM, because our data is supported by AFM analysis in Figure 4b, for the CNCs-C1a sample, but if requested we will also carry out TEM analysis and this takes a long time to send samples to other institutions.
  6. We have corrected the sample code in the experiments section, in sub-chapter 2.2.1 CNC Preparation. We have given the name an abbreviation code to make it easier to pronounce. CNCs-A0 is cellulose nano crystals prepared through a hydrolysis process with 1M HCl for 30 minutes starting from a temperature of 25 °C until it reaches 110 °C. CNCs-B0 is cellulose nano crystals prepared through a hydrolysis process with 2M HCl for 30 minutes starting from a temperature of 25 °C until reaching a temperature of 110 °C, and so on for CNCs-C0, CNCs-D0, and CNCs-E0. Meanwhile, CNCs-A1 is cellulose nano crystals prepared through a hydrolysis process with 1M HCl with a reaction time of 1 hour at a temperature of 110 °C. CNCs-B1 is cellulose nano crystals prepared through a hydrolysis process with 2M HCl with a reaction time of 1 hour at a temperature of 110 °C. CNCs-C1 is cellulose nano crystals prepared through a hydrolysis process with 3M HCl with a reaction time of 1 hour at a temperature of 110 °C, and so on for CNCs-D1 and CNCs-E1.
  7. In lines 145, 147, 327, and 329, we have replaced the abbreviation "Crl" with crystallinity.
  8. We don't understand what the size "5014" in column CNCs-A1 (Table 2) means, please explain again?
  9. Sorry, we also don't understand the meaning of "There is a duplicate title %volume for column CNCs-C1 (Table 2)", please help explain again.
  10. Figures 4 and 7 with higher resolution have been presented by us in the manuscript.
  11. We use uniform fonts in the images we send in the manuscript.
  12. Some recent references only include data.
  13. In the background we have explained several CNCs that have been made by previous researchers, but if necessary we will add them to the discussion
  14. We have omitted the perspective in Section 3.9 and added it briefly to the conclusion
  15. We have corrected it and only include data in the form of numbers
  16. We have corrected the typo, and we will check this article for language

Reviewer 2 Report

Comments and Suggestions for Authors

The manuscript reports on the use of effect of sonication-hydrothermal on acid hydrolysis to produce CNC from OPEFB. Below are few suggestions for improvement:

1. Acid hydrolysis to produce CNC has been widely explored by the researchers. Please include some discussion of green solvents, such as deep eutectic solvents and ionic liquids, in the Introduction. Highlight the research gap and objectives of study. 

2. Since the focus of the work is to study the effect of sonication-hydrothermal acid hydrolysis in production of CNC from OPEFB, the authors are advised to include the sonication parameters in experimental design. For instance, the effect of sonication power, duration and frequency. 

3. Besides, temperature has been reported to significantly affect the performance of acid hydrolysis. Please justify why it is not included in the study, or perhaps it is not included in the abstract. 

4. Please justify how the ranges of studied were made. 

5. Referring to Section 3.1 line 171 to 180 are suggested to remove. They should be presented under Methodology.

6. Referring to line 190 - 191, there is interaction effects between the investigated parameters. The authors are advised to include optimization study in the work using optimization tools to study the interaction effects and optimize the process. 

7. Referring to Table 1, there is a repetitive of sample labelling under different conditions. Please revise using different sample labelling for different reaction conditions. 

8. Please include reference(s) for the statement made in line 201 - 206. 

9. Referring to lines 214 - 221, please study the effect of sonication on the production of CNC from OPEFB, including the performance of the hydrolysis process without sonication.   

10. Section 3.2 should be enhanced by discussing the mechanism of the hydrolysis process, as there is not mechanism study found under this section. The content does not reflect the sub-heading title. 

11. Suggest to present Figure 2 as Graphical Abstract. 

12. Figure 3 is not discussed. FTIR spectra for different hydrolysis conditions should be included to study the effect of different operating conditions on hydrolysis performance. 

13. Under Section 3.5, the authors are suggested to remove lines 323 to 339, as they reflect Method rather than Results and Discussion. 

14. Referring to lines 367 - 371, the crystallinity of CNC is reported to affect the yield of CNC. Therefore, it is good to consider 2 responses during the optimization study, to optimize the operating conditions to achieve co-optimization. 

15. Suggest to remove Section 3.8 as it seems insignificant to enhance the discussion in this work. 

16. The reference list should be updated as there are no recent works (recent 5 years) found. 

17. Please proofread the manuscript to ensure it is grammatical error-free. 

Comments on the Quality of English Language

It is recommended for English language proofreading. 

Author Response

  1. Some environmentally friendly solvents include: acetic acid, but acetic acid is less effective at cracking cellulose crystal chains so it can be replaced with dilute hydrochloric acid with high hydrothermal pressure, so it can shorten the crystal chains and reduce the particle size to nanocrystals. Besides that, residual hydrochloric acid with a low concentration can be removed by diluting it with water so that it is not harmful to the environment. We have written a little of this discussion in the background in lines 52 and 68, but we will add more background as requested above. Thank You.
  2. The sonication parameters used were a sonication time of 30 minutes and a frequency of 28 KHz for all experimental variables. The role of sonication in this experiment is as swelling, causing the cellulose chains to become stressed, when in hydrothermal conditions it can easily degrade the amorphous part and then the crystal chains are fractionated into shorter and nano-sized ones. We will write this statement in sub-chapter 3.1
  3. We have included the effect of temperature in table 1, row 200, in the table you can see that the hydrolysis process with temperature differences means significant differences in the yield and percentage of crystallinity obtained, this is because if the low temperature the cellulose chains are not able to break off all the amorphous parts and crystal size still long, if the temperature is high it causes many crystal chains to degrade and decompose into glucose so that the yield becomes low. We will explain this again in sub-chapter 3.1 regarding this difference.
  4. The hydrolysis process is carried out continuously with time intervals made every (1, 2, 3, 4, 5) hours, once every 1 hour a sample is taken for analysis, and the remaining sample is continued for the next hour.
  5. We have deleted line 171 and explained it in the methodology section. Reviewer 1 asked to add a description of the CNCs-A1, CNCs-A2 codes
  6. In this study, no optimization of experimental parameters was carried out, but variations in the hydrolysis time were carried out to see the best results obtained at a hydrolysis time of several hours, so time variations were carried out from 1 to 5 hours.
  7. Referring to table 1, it is true that there is the same labeling at different hydrothermal temperature conditions, we have corrected it in the paper manuscript, where CNCs-C0 at a temperature of 100 C is changed to CNCs-C0-100, CNCs-C1 at a temperature of 100 C is changed to CNCs-C1-100. For CNCs-C0 at a temperature of 120 C it is changed to CNCs-C0-120
  8. The statement made in lines 201 - 206 is the result of a study from the research we conducted. This is reinforced by Yu et al.'s 2013 statement, we will include this reference.
  9. In this research, sonication cannot form nano-sized cellulose, but sonication only plays a role in stressing and swelling the cellulose chains so that they are easily cracked into nano-sized particles when hydrothermal.
  10. We have added a hydraulic chain breaking mechanism in section 3.2. This mechanism is in accordance with the illustration in Figure 2. Thank you for the good input.
  11. We have presented Figure 2 as a Graphic Abstract.
  12. Figure 3 is discussed globally but not in detail per spectrum. The FTIR spectrum with different hydrolysis conditions has been included in Figure 3.b, where in this image 5 FT-IR spectra have been combined, including CNCs-A1, CNCs-B1, CNCs-C1, CNCs-D1, and CNCs-E1.
  13. We have removed section 3.5, lines 323 to 339, thank you for the suggestion.
  14. The conditions of the hydrolysis process affect the resulting crystallinity, where the higher the concentration used causes the crystallinity to decrease and the yield to be lower because the higher the acid concentration can easily attack the crystal area and cause the crystal to degrade before 1 hour, this can be seen in the figure. XRd diffractogram patterns of CNCs-D1, and CNCs-E1. We have included this statement in line 367.
  15. If section 3.8 is requested to be deleted, we will delete it. The reason we discuss section 3.8 is that we want to explain that the experiments we carried out visually obtained results as shown in Figure 8.
  16. We have updated the reference list with several references (last 5 years). thank you for this suggestion.
  17. We have sent the manuscript to a proofreading service to correct the grammar.

Reviewer 3 Report

Comments and Suggestions for Authors

This manuscript reported a method called hydrochloride acid hydrolysis under sonication-hydrothermal process to produce cellulose nanocrystals from oil palm empty fruit bunch. The authors studied the effect of experimental conditions such as acid concentration, temperature, time etc. on the yield of cellulose nanocrystals. The following comments should be carefully considered by the authors.

(1) The nanostructure of the CNCs (for example, SEM or TEM) should be supported.

(2) The figures should be carefully arranged: Fig. 2b (line 266) might be Fig. 3; In Fig. 3, 3a and 3b and the (a-e) in the figure are easy to be confused, the sample name can be located above the relative line; Fig. 4a and 4b should be in the same figure or Fig. 4b numbered to be Fig. 5; (a) and (b) are not presented in Figure 6; Fig. 7 is the  screen capture figure of DLS, pleas use origin to replot the figure.

Comments on the Quality of English Language

English of the whole manuscript should be carefully checked. 

Author Response

(1) The nanostructure of CNCs in SEM analysis in Figure 4b has been explained. In Figure 7, the particle size analyzer has also explained the particle size of the CNCs

(2). It's true that it was an error, we have changed Figure 2b to 3. In Figure 3a we have changed it to 3.1 and 3b we have changed it to figure 3.2, and the sample name has been placed above the relative line;

Reviewer 4 Report

Comments and Suggestions for Authors

In this manuscript, the Authors studied the preparation of cellulose nanocrystals from oil palm empty fruit bunch via HCl acid hydrolysis to optimize the conditions. Different parameters were studied like concentration of reactants/ratios, reaction time, temperature, etc. However, this topic is not a new one. there is some literature available related to it "Hydrochloric acid hydrolysis of pulps from oil palm empty fruit bunches to produce cellulose nanocrystals" by Hastuti et al., 2018 which was not even referred.  Even a series by IOP published in 2018 to prepare CNC from OPEFB (not with HCl hydrolysis) e.g. 2018 IOP Conf. Ser.: Earth Environ. Sci. 105 012063. Those are not mentioned in the manuscript as well. Without the comparative discussion, I am unable to proceed with this article.

Author Response

We have taken several references and comparisons with related articles such as Yu et al. 2013 and Filson et al. 2013. These two articles were the best research results at that time. However, we will include your suggestions in this article for comparison with the latest article

Round 2

Reviewer 1 Report

Comments and Suggestions for Authors

The concerns are now addressed, so the revised manuscript can be accept for publication.

Author Response

Thank you for accepting this manuscript. Thank you also for providing a lot of input on my research manuscript, so that the writing has better progress and can be read by the general public and can become a reference for research experts. Hopefully this manuscript will be accepted in the Polymers journal

Reviewer 2 Report

Comments and Suggestions for Authors

All comments have been addressed.

Comments on the Quality of English Language

English proofreading is recommended.

Author Response

Thank you for accepting this manuscript. Thank you also for providing a lot of input on my research manuscript, so that the writing has better progress and can be read by the general public and can become a reference for research experts. Hopefully this manuscript will be accepted in the Polymers journal.
We are currently carrying out English proofreading at an official institution so that the grammar of this manuscript is better.

Reviewer 3 Report

Comments and Suggestions for Authors

1. Please check the figure numbers of Figure 3.1 and Figure 3.2, Figure 4.a and Figure 4.b, as others are named in the form of "Figure 1" 

2. In Figure 6, (a) and (b) are lost. 

3. In same figure caption, "Typical ..." was used.  Why the word "typical" was used?

4. Please check the Y axis of Figure 3.1 "Transmitance" which should be "Transmittance" and please make the two FTIR images in the same format.

Author Response

  1. We have corrected Figure 4.a and Figure 4.b to become Figure 4.1 and Figure 4.2, and we have highlighted them in yellow in the text
  2. Figure 6, (a) and (b) we have separated the revisions into Fig. 6.1 and 6.2
  3. There is indeed a misunderstanding of the word typical, but what we mean is that we want to explain that the FT-IR spectrum is a typical characteristic of cellulose and CNCs. However, we have removed it as we highlighted in Figures 3.1 and 3.2
  4. We have corrected the Y axis in Figure 3.1 and we have made the writing format uniform, we have highlighted it in the image caption

Reviewer 4 Report

Comments and Suggestions for Authors

Unfortunately, Author did not take comment seriously. One previous literature, (Hastuti et al., 2018) reported similar works. Authors just mentioned that. It is required to mention about the difference with current manuscript from that one by comparing all the data side-by-side. I attached that  in the attachment.

Author Response

We have compared several parameters between Hastuti's research and our research, which we have highlighted, including:
1. Our experimental hydrolysis process with (1, 2, 3, 4, 5) M hydrochloric acid at 1 hour reaction time respectively under sonication-hydrothermal process at 110°C showed a high yield and crystallinity obtained, with yield 52.2; 79.09; 74.8; 72.9 and 60.5% respectively. (line 225).
Compared with research by Hastuti (2018). The yields of the CNC-A, B, and C from the three pulps, i.e., pulp-A, -B, and -C, were 21, 18, and 19%, respectively. (lines 229- 236).

2. The crystallinity of the resultant CNCs by Hastuti (2018) was also lower compared to our research, namely CNC-A, B, and C were 65, 60, and 53%, respectively. Meanwhile, in our research, CNC-A1, CNC-B1 and CNC-C1 were 70.63, 75.87 and 78.59% respect-atively. (lines 380-383).

3. In our study the Tmax of CNC-A1, CNC-B1, and CNC-C1 were 326.72, 340.15, and 339.82, respectively. (lines 459-464).
The Tmax in Hastuti's research showed good thermal stability for CNC-A, CNC-B, and CNC-C respectify 358.5, 359.2 and 346.5 oC (Hastuti et al., 2018). (lines 472-473).
Hastuti also reported that EFB has higher thermal stability, namely 347–359 °C as the maximum degradation temperature, compared to wood CNC made through sulfuric acid hydrolysis (15 and 311 °C respectively) (Hastuti et al., 2018). (lines 474-476).

Round 3

Reviewer 4 Report

Comments and Suggestions for Authors

I am fine with this version.